# Improvement in Crystallization, Thermal, and Mechanical Properties of Flexible Poly(L-lactide)-*b*-poly(ethylene glycol)-*b*-poly(L-lactide) Bioplastic with Zinc Phenylphosphate

**DOI:** 10.3390/polym16070975

**Published:** 2024-04-03

**Authors:** Kansiri Pakkethati, Prasong Srihanam, Apirada Manphae, Wuttipong Rungseesantivanon, Natcha Prakymoramas, Pham Ngoc Lan, Yodthong Baimark

**Affiliations:** 1Biodegradable Polymers Research Unit, Department of Chemistry and Centre of Excellence for Innovation in Chemistry, Faculty of Science, Mahasarakham University, Mahasarakham 44150, Thailand; kansiri.p@msu.ac.th (K.P.); prasong.s@msu.ac.th (P.S.); apirada.m@msu.ac.th (A.M.); 2Scientific Instrument Academic Service Unit, Faculty of Science, Mahasarakham University, Mahasarakham 44150, Thailand; 3National Metal and Materials Technology Centre (MTEC), 114 Thailand Science Park (TSP), Phahonyothin Road, Khlong Nueng, Khlong Luang, Pathum Thani 12120, Thailand; wuttir@mtec.or.th (W.R.); natchap@mtec.or.th (N.P.); 4Faculty of Chemistry, University of Science, Vietnam National University-Hanoi, 19 Le Thanh Tong Street, Phan Chu Trinh Ward, Hoan Kiem District, Hanoi 10000, Vietnam; phamngoclan49@gmail.com

**Keywords:** poly(lactic acid), poly(ethylene glycol), block copolymer, nucleating agent, thermal stabilizer, reinforcing filler

## Abstract

Poly(L-lactide)-*b*-poly(ethylene glycol)-*b*-poly(L-lactide) (PLLA-PEG-PLLA) shows promise for use in bioplastic applications due to its greater flexibility over PLLA. However, further research is needed to improve PLLA-PEG-PLLA’s properties with appropriate fillers. This study employed zinc phenylphosphate (PPZn) as a multi-functional filler for PLLA-PEG-PLLA. The effects of PPZn addition on PLLA-PEG-PLLA characteristics, such as crystallization and thermal and mechanical properties, were investigated. There was good phase compatibility between the PPZn and PLLA-PEG-PLLA. The addition of PPZn improved PLLA-PEG-PLLA’s crystallization properties, as evidenced by the disappearance of the cold crystallization temperature, an increase in the crystallinity, an increase in the crystallization temperature, and a decrease in the crystallization half-time. The PLLA-PEG-PLLA’s thermal stability and heat resistance were enhanced by the addition of PPZn. The PPZn addition also enhanced the mechanical properties of the PLLA-PEG-PLLA, as demonstrated by the rise in ultimate tensile stress and Young’s modulus. We can conclude that the PPZn has potential for use as a multi-functional filler for the PLLA-PEG-PLLA composite due to its nucleating-enhancing, thermal-stabilizing, and reinforcing ability.

## 1. Introduction

Biodegradable bioplastics have been widely investigated because they are considered environmentally friendly, sustainable, and renewable, and they also reduce pollution of wastes of conventional petroleum-based plastics. Poly(L-lactide) (PLLA) derived from renewable resources, such as starch-rich crops and sugarcane, is an important biodegradable bioplastic alternative to petroleum-based plastics because of its good biodegradability, biocompatibility, and processability, and also because it is the cheapest biodegradable bioplastic [1,2,3]. PLLA has been applied for use in biomedical applications [4,5,6], agriculture [7], sport [8], and packaging [9,10,11,12]. However, the low flexibility of PLLA has restricted its practical use and wider applications [3,13].

In order to increase PLLA’s flexibility by increasing its chain mobility, an extensive range of plasticizers has been blended with PLLA [14,15,16]. The selection of a good plasticizer for PLLA has considered a number of factors, including plasticizer efficiency, non-toxicity, good miscibility, and durability. Among all of the plasticizers, poly(ethylene glycol) (PEG) was the most effective for PLLA because it is non-toxic, it has good miscibility, and it has the ability to accelerate PLLA’s hydrolytic degradation [16]. Nevertheless, the primary drawback has been identified as phase separation and PEG migration from PLLA matrices [17,18,19], which has a direct impact on the stability and durability of the PLLA/PEG blends throughout storage and use [20,21].

High-molecular-weight PLLA-*b*-poly(ethylene glycol)-*b*-PLLA (PLLA-PEG-PLLA) triblock copolymers have more flexibility than PLLA [22,23]. This is due to the PEG middle-blocks acting as plasticizing sites to improve the chain mobility of PLLA end-blocks and decrease the glass transition temperature of PLLA end-blocks from about 60 °C to about 30 °C [23]. Moreover, the crystallization properties of the PLLA end-blocks were improved by the plasticizing effect of PEG middle-blocks. However, the melt strength of PLLA-PEG-PLLA is too low because of the plasticization effect of PEG middle-blocks. The melt strength of the PLLA-PEG-PLLA can be improved for conventional melt processing, such as injection molding, by reacting with a chain extender to form branching structures through post-treatment [23] and in situ copolymerization [24]. However, the obtained chain-extended PLLA-PEG-PLLA showed a slower crystallization rate and lower heat-resistant properties than the non-chain-extended PLLA-PEG-PLLA because the branching structures of chain-extended PLLA-PEG-PLLA inhibited the crystallization of PLLA end-blocks, thereby limiting its wider applications [25].

Depending on the thermal history of PLLA, it can be either amorphous or semicrystalline. PLLA will become highly amorphous upon quenching it from the melt phase (for example, during the extrusion and injection processes) [26]. Addition of a nucleating agent to the PLLA is an effective method for improving the crystallization of PLLA during its processing. Crystallization at high temperature can occur because the surface free energy barrier for nucleation is lowered upon cooling. Many nucleating agents have been used to enhance the crystallization properties and to increase the crystallinity of PLLA [16]. Examples of these nucleating agents are talcum [16,27,28], zinc phenylphosphate (PPZn) [16,27,29,30,31], stereocomplex PLA [16,27,32], and starch [16,33]. Highly crystalline PLLA exhibits good heat-resistance properties due to the improved stiffness [16,34]. It has been reported that PPZn is a more effective nucleating agent than both talcum powder and PLA stereocomplex [16].

For chain-extended PLLA-PEG-PLLA, various nucleating agents, such as talcum [35], calcium carbonate [24], and native starch [33], have been investigated to improve the crystallization properties of PLLA end-blocks of PLLA-PEG-PLLA. Among these nucleating agents, talcum was an effective nucleating agent, improving both the crystallization and heat-resistant properties of the chain-extended PLLA-PEG-PLLA. To the best of our knowledge, the nucleation effects of PPZn on crystallization and heat-resistant properties of chain-extended PLLA-PEG-PLLA have not been reported so far. Thus, the objective of this work is to study the effect of PPZn on non-isothermal and isothermal crystallization, heat resistance, mechanical properties, and PLLA thermal stability of the chain-extended PLLA-PEG-PLLA.

## 2. Materials and Methods

### 2.1. Materials

The chain-extended PLLA-PEG-PLLA was synthesized through ring-opening polymerization of L-lactide monomer, as described in our previous works [24,36]. PEG with molecular weight of 20,000 and stannous octoate were used as the initiating system. The number-averaged molecular weight (*M_n_*) and dispersity (*Ð*) obtained from gel permeation chromatography (GPC) of the obtained PLLA-PEG-PLLA were 108,500 and 2.2, respectively. PPZn was synthesized from phenylphosphonic acid (98%, Acros Organics, Geel, Belgium) and zinc chloride (ZnCl_2_, 98%, Acros Organics, Geel, Belgium) according to the literature [29]. Figure 1a shows a SEM image of the obtained PPZn powder. The average particle size of 100 PPZn particles determined from the SEM image using an ImageJ program was 4.12 ± 1.60 µm. The particle size distribution of PPZn powder is presented in Figure 1b. The thermal decomposition of PPZn powder determined from the thermogravimetric (TG) thermogram was in the range of 550–700 °C, as shown in Figure 1c. The residue weight at 800 °C of PPZn powder was about 73%. The XRD pattern of PPZn powder is illustrated in Figure 1d, with XRD peaks at 6.5°, 12.6°, and 18.5°.

### 2.2. Preparation of PLLA-PEG-PLLA/PPZn Composites

PLLA-PEG-PLLA and PPZn were dried in a vacuum oven at 50 °C overnight before melt mixing with a HAAKE internal mixer Polylab OS System (Waltham, MA, USA) at 190 °C for 6 min. A rotor speed of 100 rpm was used. The PLLA-PEG-PLLA composites with PPZn contents of 0.5, 1, 2, and 4 %wt were investigated. The ground particles of composites were dried in a vacuum oven at 50 °C overnight before compression molding with a Carver compression molding machine Auto CH Carver (Wabash, IN, USA) at 190 °C for 3 min without compression force, followed by a 5 MPa compression force for 2 min. The obtained film was immediately cooled for 3 min under 5 MPa compression force with water-cooled plates. The thicknesses of the obtained films were in the range of 0.2–0.3 mm.

### 2.3. Characterization of PLLA-PEG-PLLA/PPZn Composites

Differential scanning calorimetry (DSC) was conducted on a PerkinElmer DSC Pyris Diamond (Waltham, MA, USA) under a nitrogen atmosphere. For non-isothermal analysis, the samples were maintained at 200 °C for 3 min to erase the previous thermal history before quickly quenching to 0 °C with a cooling rate of 100 °C/min. The samples were then scanned from 0 °C to 200 °C with a heating rate of 10 °C/min for DSC heating scans. For the DSC cooling scan, the samples were heated at 200 °C for 3 min to erase the previous thermal history before scanning from 200 °C to 0 °C with a cooling rate of 10 °C/min. The degree of crystallinity of the samples, as determined by the DSC (*DSC-X_c_*) of PLLA crystallites, was calculated using the following equation:*DSC-X_c_* (%) = [(Δ*H_m_* − Δ*H_cc_*)/(93.6 × *W_PLLA_*)] × 100(1)
where Δ*H_m_* and Δ*H_cc_* are the melting and cold-crystallization enthalpies, respectively. For 100%*DSC-X_c_* of PLLA, the Δ*H_m_* value is 93.6 J/g [37]. *W_PLLA_* is the weight fraction of PLLA.

For isothermal analysis, the samples were isothermally crystallized at 120 °C according to the literature [38], as follows. The samples were first heated at 200 °C for 3.0 min, cooled to 120 °C at a rate of 50 °C/min, and subsequently isothermal scanned at 120 °C until the crystallization process was completed. The time needed to achieve 50% of the final crystallinity is known as the crystallization half-time (*t*_1/2_).

Wide-angle X-ray diffractometry (XRD) was performed on a Bruker Corporation XRD D8 Advance (Karlsruhe, Germany) to determine crystalline structures with CuKα radiation at 40 kV and 40 mA. The scan speed was 3°/min. The degree of crystallinity as determined according to the XRD (*XRD-X_c_*) of PLLA crystallites was calculated using the following equation:*XRD-X_c_* (%) = [(*A_c_*)/(*A_c_* + *A_a_*)] × 100(2)
where *A_c_* is the peak area of PLLA crystallites and *A_a_* is the halo area of the amorphous phase.

Thermogravimetric analysis (TGA) was performed on a TA Instruments TGA SDT Q600 (New Castle, DE, USA) from room temperature to 800 °C with a heating rate of 20 °C/min under nitrogen flow with a rate of 100 mL/min.

Dynamic mechanical analysis (DMA) was conducted on a TA Instrument DMA Q800 DMA (New Castle, DE, USA) with a tension mode from 30 °C to 150 °C at a heating rate of 2 °C/min. The scan amplitude was 10 µm, and the scanning frequency was 1 Hz.

The heat resistance of the composite films was investigated by testing the dimensional stability to heat at 80 °C under a 200 g load for 30 s [39,40]. The initial gauge length of the films was 20 mm. The dimensional stability to heat of the film samples was calculated using the following equation. The average value was obtained from five different determinations of each film sample.
Dimensional stability to heat (%) = [initial gauge length/final gauge length] × 100(3)

Cryo-fractured surfaces of the composite films were analyzed to observe the phase separation between the PLLA-PEG-PLLA matrix and the PPZn particles through scanning electron microscopy (SEM) on a JEOL JSM-6460LV SEM (Tokyo, Japan) at 15 kV. Before scanning, the film samples were gold sputter coated.

A tensile test was performed on a LY-1066B universal testing machine (Dongguan Liyi Environmental Technology Co., Ltd., Dongguan, China) according to the ASTM D882 at 25 °C with a crosshead speed of 50 mm/min. The initial gauge length was 50 mm, and the load cell was 100 kg. Each film sample was tested with a minimum of five determinations.

The film’s opacity was determined using a Thermo Scientific Genesys 20 visible spectrophotometer (Loughborough, UK) and calculated with the following equation [41]:Opacity (mm^−1^) = *A*_600_/*X*(4)
where *A*_600_ is the absorbance of the film at 600 nm and *X* is the thickness of the film sample (mm).

## 3. Results

### 3.1. Thermal Transition Properties

The PLLA-PEG-PLLA composites with different PPZn contents were prepared to compare the nucleating effect of PPZn on the non-isothermal and isothermal DSC scans of the PLLA-PEG-PLLA matrix, as shown in Figure 2 and Figure 3, respectively. In Figure 2a, the glass transition (*T_g_*), cold crystallization (*T_cc_*), and melting (*T_m_*) temperatures are seen on the DSC heating thermograms. In Figure 2b, the crystallization temperature (*T_c_*) is seen on the DSC cooling thermograms. The DSC results of both the DSC heating and cooling scans are summarized in Table 1. The *T_g_*, *T_cc_*, and *T_m_* values of PLLA end-blocks of pure PLLA-PEG-PLLA were 31 °C, 81 °C, and 152 °C, respectively. The *T_g_* and *T_cc_* values of pure PLLA-PEG-PLLA disappeared when the PPZn was incorporated. This may be due to the glassy-to-rubbery transition in amorphous regions of PLLA end-blocks after quenching from 200 °C to 0 °C being difficult to detect by DSC for PLLA-PEG-PLLA composites because the free volume of the polymer chains was reduced for high-crystallinity polymers, which leads to restricting the motion of polymer chains in amorphous regions [42]. The disappearance of a *T_cc_* peak of the PLLA-PEG-PLLA matrix indicates that the composites underwent complete crystallization during quenching from 200 °C to 0 °C [38,43,44]. It should be noted that lower-temperature shoulder *T_m_* peaks were also detected when the PPZn was incorporated, which suggests that imperfect crystals of PLLA end-blocks were formed [29].

Degrees of crystallinity assessed from DSC (*DSC-X_c_*) calculated from Equation (1) are also reported in Table 1. The *DSC-X_c_* value of pure PLLA-PEG-PLLA was 12.3% due to the crystallites of PLLA end-blocks. When the 0.5 %wt PPZn was incorporated, the *DSC-X_c_* value of the PLLA-PEG-PLLA composite was dramatically increased up to 42.2%. This suggests that the PPZn acted as an effective nucleating agent for PLLA-PEG-PLLA. The *DSC-X_c_* values slightly increased as the PPZn contents increased higher than 0.5 %wt. It should be noted that the *DSC-X_c_* value of PLLA-PEG-PLLA composite containing 4 %wt talcum was 48.5% in our previous work [35], whereas the PLLA-PEG-PLLA composite containing 4 %wt PPZn in this work was 49.6%. This suggests the PPZn had a better nucleating effect than the talcum when comparing data reported in the literature [16].

From DSC cooling thermograms, the *T_c_* peaks of the PLLA-PEG-PLLA matrix dramatically shifted from 107 °C to 118 °C when the 0.5 %wt PPZn was incorporated, thus supporting the conclusion that added PPZn enhanced crystallization of PLLA-PEG-PLLA through a nucleating effect [45,46,47]. The *T_c_* peaks slightly shifted to higher temperatures as the PPZn contents were higher than 0.5 %wt.

Furthermore, the crystallization properties of the composites were studied as derived from the crystallization half-time (*t*_1/2_) that was obtained from isothermal crystallization curves at 120 °C, as shown in Figure 3a. The 50% relative crystallinity of polymer samples was obtained during isothermal scans at the time of *t*_1/2_, as shown in Figure 3b. The resulting *t*_1/2_ values are summarized in Table 2. The *t*_1/2_ value of pure PLLA-PEG-PLLA was 2.18 min. The time was significantly decreased to 0.75 min when 0.5 %wt PPZn was added, thereby supporting the effective nucleating effect of added PPZn. The *t*_1/2_ values steadily decreased as the PPZn contents increased higher than 0.5 %wt. The crystallization kinetics of the composites were investigated using the following Avrami equation [38,48]:1 − *X_t_* = exp(−*kt^n^*)(5)
where *X_t_* is the relative crystallinity as a function of time, *t* is the crystallization time, *n* is the Avrami exponent, and *k* is the crystallization rate constant.

The resulting Avrami parameters (*n* and *k* values) determined from log[−ln(1 − *X_t_*)] versus log(*t*) curves are also reported in Table 2. Graphs with good linear regression were obtained. All R^2^ were higher than 0.99. Pure PLLA-PEG-PLLA appeared to have the slowest crystallization, as indicated by its highest *n* and lowest *k* values [48,49]. The *n* values of the composites steadily decreased and the *k* values significantly increased as the PPZn content increased, supporting a conclusion that the addition of PPZn improved the crystallization properties of PLLA-PEG-PLLA matrices [48,49]. All of the *n* values were higher than 2.0, suggesting a heterogeneous nucleation effect [50]. The DSC results from both non-isothermal and isothermal scans justified a conclusion that the PPZn acted as a good nucleating agent to increase the crystallinity of PLLA-PEG-PLLA.

### 3.2. Thermal Decomposition Behaviors

The thermal decomposition behaviors of the composites were determined from thermogravimetric (TG) and derivative TG (DTG) thermograms as well as their expanded thermograms, as shown in Figure 4. From Figure 4a, the pure PLLA-PEG-PLLA indicated by the black curve had two-step thermal decompositions of PLLA end-blocks (200–375 °C) and PEG middle-blocks (375–450 °C) [23,24]. All of the composite samples also had two-step thermal decompositions on TG thermograms similar to the pure PLLA-PEG-PLLA. As would be expected, the residue weight at 800 °C of the composites increased as the PPZn content increased (see Table 3). This is because the PPZn did not completely thermally decompose at 800 °C, as shown in Figure 1c.

The DTG thermograms in Figure 4b exhibited peaks of temperature at the maximum decomposition rate (*T_d,max_*) for PLLA (*PLLA-T_d,max_*) and for PEG (*PEG-T_d,max_*). The results of the *T_d,max_* values are also summarized in Table 3. For pure PLLA-PEG-PLLA, the *PLLA-T_d,max_* peak was at 323 °C and the *PEG-T_d,max_* peak was at 416 °C. The *PLLA-T_d,max_* peaks of the composites were in the range of 328–329 °C, which were higher temperatures than that of the pure PLLA-PEG-PLLA (323 °C) when the PPZn was incorporated. The higher *PLLA-T_d,max_* values of the composites might indicate increased thermal stability of the composites compared to pure PLLA-PEG-PLLA. This may be due to a good heat transfer from the PLLA-PEG-PLLA matrices to the high-heat-stability PPZn that improved the thermal stability of the PLLA-PEG-PLLA, attributed to good phase adhesion between the matrix and filler [51]. However, the addition of PPZn did not significantly change the *PEG-T_d,max_* peaks. The *PEG-T_d,max_* peaks were in the range of 415–418 °C. This may be due to the phases of PLLA end-blocks being completely thermally decomposed at around 380 °C [see Figure 4b]. Then, almost all of the PLLA-PEG-PLLA matrices would have been destroyed, and there would be no more heat transfer from the PLLA-PEG-PLLA matrices to the PPZn. From the TGA results, a good heat transfer between the PLLA-PEG-PLLA matrix and PPZn also suggested good phase compatibility between them that SEM analysis would subsequently verify. In addition, Yang et al. have reported that the thermal stability of PLLA was improved with increasing crystallinity of PLLA [52].

### 3.3. Crystalline Structures

The crystalline structures of film samples were investigated from XRD patterns, as shown in Figure 5. The pure PLLA-PEG-PLLA in Figure 5a had a broad XRD peak at 16.9° due to PLLA end-block crystallites [47,50]. For the composite films in Figure 5b–e, the XRD peaks of added PPZn were detected at 6.5°, 12.6°, and 18.5°. It was found that the intensities of XRD peaks at 16.9° of composite films steadily increased as the PPZn contents increased. In addition, the XRD peaks appeared at 14.8°, 19.1°, and 22.5° of the composite film containing 4 %wt PPZn due to PLLA crystallites [38,44]. Thus, the addition of PPZn did not change the crystalline structures of the PLLA end-blocks.

The *XRD-X_c_* value of pure PLLA-PEG-PLLA film calculated from Equation (2) was 7.7%. The *XRD-X_c_* values of film samples increased when the added PPZn content was increased. The *XRD-X_c_* values were 9.4%, 10.4%, 15.6%, and 28.1% for the composite films containing PPZn contents of 0.5 %wt, 1 %wt, 2 %wt, and 4 %wt, respectively. The XRD results supported the conclusion that the PPZn acted as a nucleating agent according to the above DSC results. The difference between the *DSC-X_c_* and *XRD-X_c_* values could be due to the differences in crystallization conditions during the cooling process of the samples [27,53]. In addition, the film samples used in the XRD test were compressed and cooled to limit the mobility of the polymer chains for crystallization.

### 3.4. Thermo-Mechanical Properties

The thermo-mechanical properties of PLLA and PLLA-PEG-PLLA are related to their heat resistance properties, which have been widely determined through DMA analysis [27,40,43,45,54,55,56,57,58,59]. The variation in the storage modulus of PLLA-PEG-PLLA as a function of temperature is presented in Figure 6. Pure PLLA-PEG-PLLA, indicated by the black curve, first displayed a drop curve of storage modulus in the range of 30–70 °C, suggesting that it had low stiffness [40] because of the rubbery character and low crystallinity of PLLA-PEG-PLLA. At higher temperatures, this was followed by an increase in storage modulus again because of the cold crystallization of the PLLA end-blocks [27,40]. This indicates that pure PLLA-PEG-PLLA had poor heat resistance because the pure PLLA-PEG-PLLA film had low crystallinity, as described in the above XRD results (*XRD-X_c_* of pure PLLA-PEG-PLLA is 7.7%). It has been reported that the phases of PLLA crystallites induce high storage modulus and increase heat-resistant properties [54,55,56,59].

The cold-crystallization effect of DMA curves from Figure 6 disappeared when the PPZn content was increased up to 1 %wt (see blue curve), indicating that the crystallinity of composite film was high enough to erase the cold-crystallization behavior [33]. The lowest storage modulus of pure PLLA-PEG-PLLA in the temperature range of 30–120 °C was only 29 MPa. The lowest storage moduli of the composites containing PPZn contents of 0.5, 1.0, 2.0, and 4.0 %wt were 70, 91, 125, and 209 MPa, respectively. The lowest storage moduli of the composites increased as the PPZn content increased. From the DMA results, it was concluded that the addition of PPZn increased the film stiffness of the PLLA-PEG-PLLA matrices during the DMA heating scan, enabling it to resist film deformation and improve film heat resistance. This may be explained by the fact that the addition of PPZn increased the crystallinity of PLLA-PEG-PLLA, thus maintaining the storage modulus of composite films during the DMA heating scan [19,40,43,45].

### 3.5. Dimensional Stability to Heat

The dimensional stability to heat of film samples at 80 °C under the load of 200 g for 30 s was also used to examine their heat-resistant properties. The film samples with higher heat resistance exhibited shorter final lengths and higher values of dimensional stability to heat [39,40]. Images of pure PLLA-PEG-PLLA and composite films before and after dimensional stability testing are displayed in Figure 7. It can be seen that the pure PLLA-PEG-PLLA showed the longest film extension, indicating that it had the lowest heat resistance. The film extension significantly decreased as the PPZn content increased.

The values of dimensional stability to heat were calculated from Equation (3) and compared in bar graphs, as shown in Figure 8. The pure PLLA-PEG-PLLA had the lowest dimensional stability to heat at about 28.3% because it had the lowest crystallinity. The values of dimensional stability to heat steadily increased as the PPZn content increased. These values were about 47.6%, 58.7%, 82.8%, and 96.1% for PPZn content of 0.5, 1, 2, and 4 %wt, respectively. This may be explained by the fact that the dimensional stability to heat of film samples strongly depended on the *XRD-X_c_* values of film samples [39,40,55]. The crystalline phases of PLLA end-blocks of PLLA-PEG-PLLA matrices improved film stiffness to resist film extension at 80 °C. The results supported the conclusion that the addition of PPZn improved the heat resistance of composited films according to the above DMA analysis.

### 3.6. Phase Compatibility

The phase compatibility between the PLLA-PEG-PLLA matrix and PPZn of the composite films was investigated based on their cryo-fractured surfaces, as shown in Figure 9. The pure PLLA-PEG-PLLA film in Figure 9a showed rougher surfaces, suggesting that the film was flexible. Some small, white, needle-shaped streaks were also observed on the fractured surfaces. This could be because the PLLA-PEG-PLLA matrix was somewhat stretched before breaking in the cryo-fracture step. These tiny, white needles were also found on the fractured surfaces of all of the PLLA-PEG-PLLA/PPZn composite films, indicating that they were also flexible. Some PPZn particles of composite films in Figure 9b–e are indicated by white circles. It can be seen that the PPZn particles were well-distributed and dispersed on the PLLA-PEG-PLLA matrices, suggesting that the PPZn particles and PLLA-PEG-PLLA matrix had good phase compatibility.

### 3.7. Tensile Properties

Stress–strain curves from tensile testing were used to characterize the mechanical properties of the composite films, as shown in Figure 10. The tensile results are summarized in Table 4 and are clearly compared in bar graph types, as shown in Figure 11. The pure PLLA-PEG-PLLA film had an ultimate tensile stress of 13.5 MPa, a strain at break of 125%, and a Young’s modulus of 190 MPa. All composite films, including the pure PLLA-PEG-PLLA film, exhibited a yield point. This result confirms the conclusion of the above SEM study that the composite films were flexible. With increased PPZn content, the ultimate tensile stress and Young’s modulus of the composite films significantly increased, and the strain at break steadily decreased. The tensile results suggested that the PPZn acted as a reinforcing agent for PLLA-PEG-PLLA. This may be due to the crystallinity of composite films increasing as the PPZn was incorporated, as explained in the above XRD analysis. The crystalline phases could act as physical cross-linking sites to improve the tensile stress and Young’s modulus of the flexible polymers [43,52]. In addition, the increase in tensile stress of PLLA-PEG-PLLA/PPZn films demonstrated the effective interaction between PLLA-PEG-PLLA and PPZn [60], and the reinforcing effect of PPZn subsequently restricted the chain mobility and deformation of PLLA-PEG-PLLA.

It was shown in our previous study [28] that the addition of 2 %wt talcum also improved the ultimate tensile stress of PLLA-PEG-PLLA. The talcum became aggregated when its content was higher than 2 %wt, reducing the tensile properties of the PLLA-PEG-PLLA composite films. This suggested poor phase compatibility between the PLLA-PEG-PLLA and talcum. However, this study showed that the ultimate tensile stress steadily increased with increasing the PPZn-content up to 4 %wt. This suggested a better phase compatibility of the PLLA-PEG-PLLA with PPZn than with talcum. Thus, PPZn can be considered a good reinforcement for PLLA-PEG-PLLA.

### 3.8. Film’s Opacity

The opacity of pure PLLA-PEG-PLLA film calculated from Equation (4) was 0.402 mm^−1^, as also reported in Table 4. It was found that the opacity of film samples increased when the added PPZn content was increased. The composite films had higher opacity compared with the pure PLLA-PEG-PLLA film, as shown in Figure 12. However, the words covered by the composite films were still clearly visible, and they were legible. It has been reviewed that the polymer film’s opacity increases as the crystallinity of polymers increases [61].

## 4. Conclusions

This study aimed to understand the role and behavior of zinc phenyl phosphate (PPZn) when incorporated with the poly(L-lactide)-*b*-poly(ethylene glycol)-*b*-poly(L-lactide) (PLLA-PEG-PLLA) flexible bioplastic. The addition of the PPZn to PLLA-PEG-PLLA effectively improved the crystallization properties, thermal stability, heat resistance, and tensile properties of PLLA-PEG-PLLA matrices as follows. The *T_cc_* peaks of the PLLA-PEG-PLLA/PPZn composites disappeared, the *T_c_* peaks shifted to higher temperature, the crystallinity increased, and the *t*_1/2_ decreased as the PPZn content increased, suggesting that the PPZn acted as a good nucleating agent. The PLLA-Td,max of the PLLA-PEG-PLLA composites shifted to a higher temperature with the PPZn contents, indicating that the PPZn acted as a thermal stabilizer. Increasing the stiffness and the dimensional stability to heat of PLLA-PEG-PLLA composites by adding PPZn is conductive to improving the heat resistance of PLLA-PEG-PLLA matrices because the crystallinity increases. The PLLA-PEG-PLLA/PPZn composites showed good phase compatibility, which led to enhanced crystallization properties and thermal stability of the PLLA-PEG-PLLA matrices. The ultimate tensile stress and Young’s modulus of PLLA-PEG-PLLA matrices steadily increased as the PPZn content increased, suggesting that the PPZn acted as a reinforcing filler. The film’s opacity increased with the PPZn content. Thus, PPZn can be used as an efficient multi-functional filler to develop the PLLA-PEG-PLLA’s properties for widespread bioplastic applications.

## Figures and Tables

**Figure 1 polymers-16-00975-f001:**
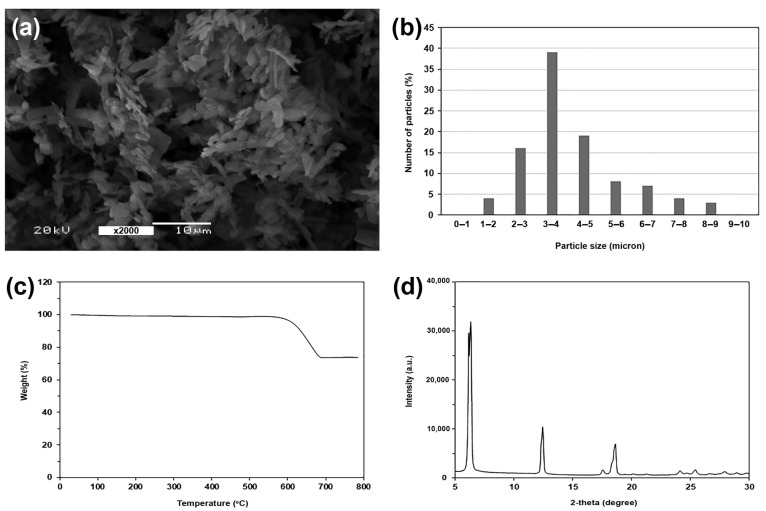
(**a**) SEM image, (**b**) particle size distribution, (**c**) TG thermogram, and (**d**) XRD pattern of PPZn powder.

**Figure 2 polymers-16-00975-f002:**
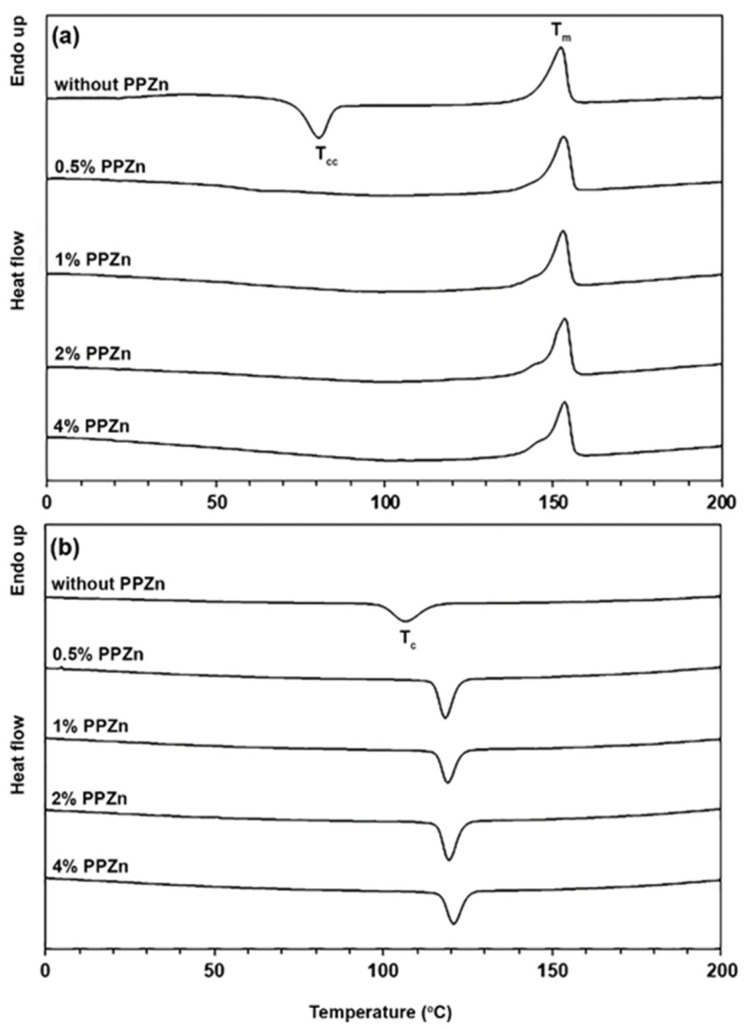
DSC thermograms of (**a**) heating scans and (**b**) cooling scans of PLLA-PEG-PLLA/PPZn composites with various PPZn contents.

**Figure 3 polymers-16-00975-f003:**
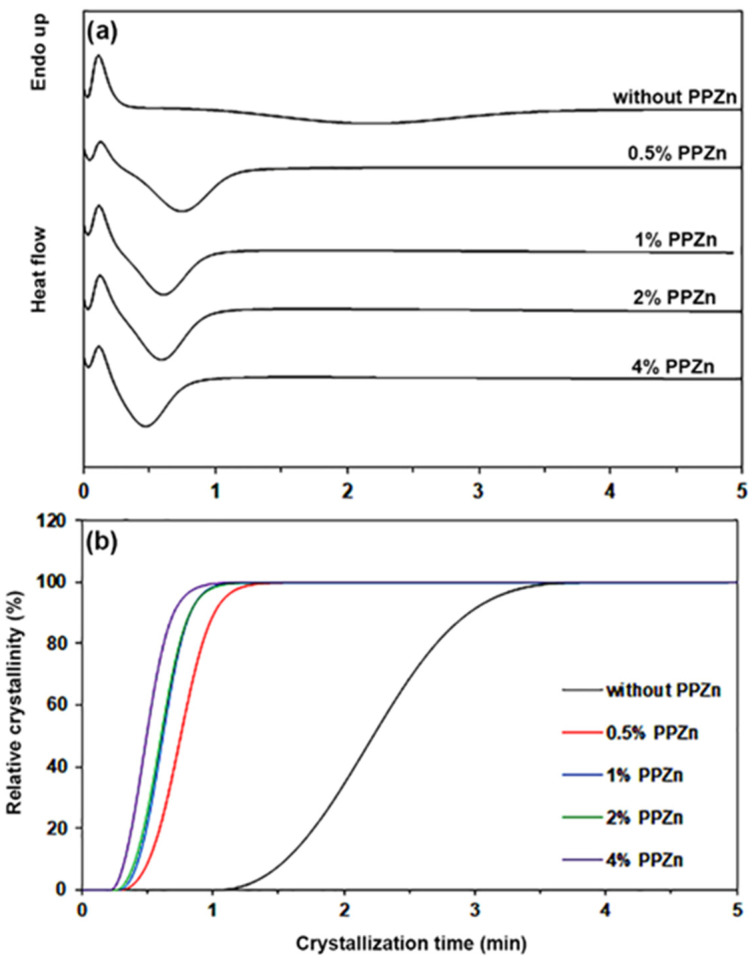
(**a**) Isothermal crystallization curves at 120 °C and (**b**) relative crystallinity–crystallization time curves of PLLA-PEG-PLLA/PPZn composites with various PPZn contents.

**Figure 4 polymers-16-00975-f004:**
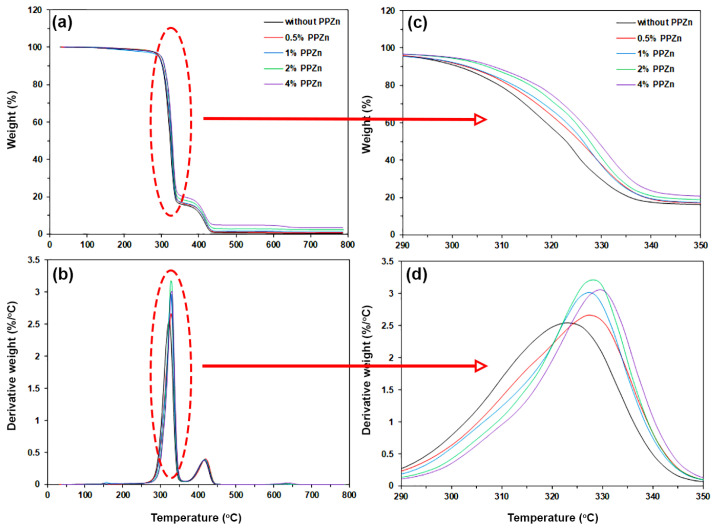
(**a**) TG and (**b**) DTG thermograms of PLLA-PEG-PLLA/PPZn composites with various PPZn contents as well as (**c**) expanded TG and (**d**) expanded DTG thermograms in decomposition region of PLLA end-blocks.

**Figure 5 polymers-16-00975-f005:**
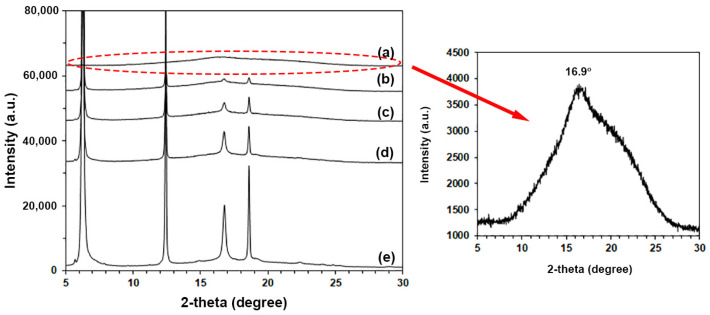
XRD patterns of PLLA-PEG-PLLA/PPZn composites (a) without PPZn and with PPZn contents of (b) 0.5 %wt, (c) 1 %wt, (d) 2 %wt, and (e) 4 %wt.

**Figure 6 polymers-16-00975-f006:**
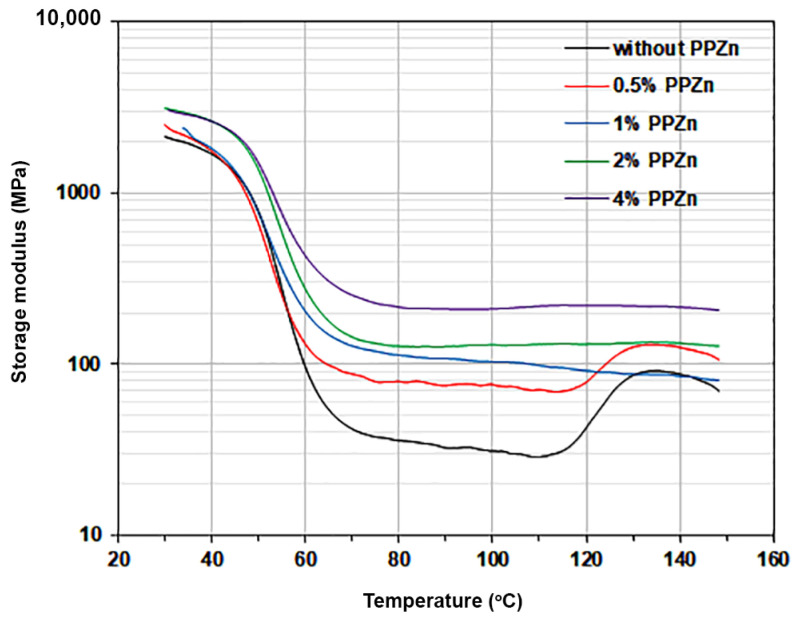
Storage modulus as a function of temperature of PLLA-PEG-PLLA/PPZn composite films with various PPZn contents.

**Figure 7 polymers-16-00975-f007:**
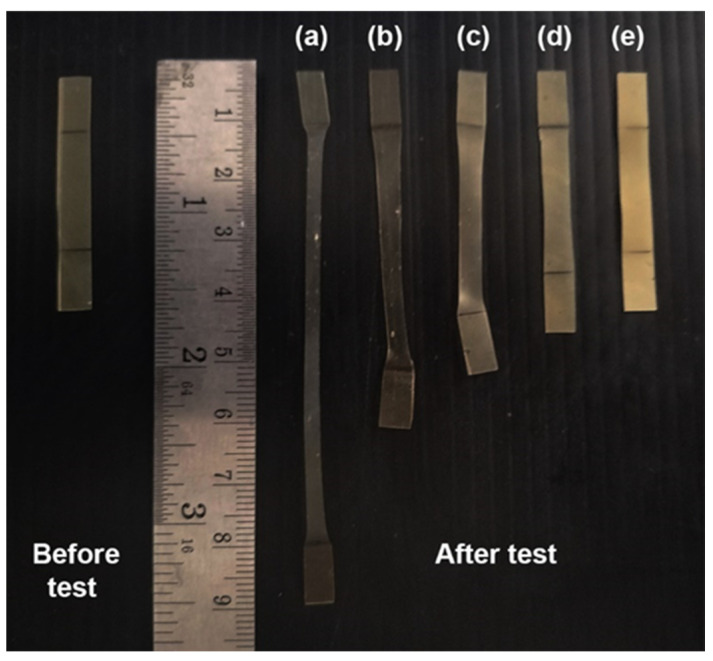
Photographs of PLLA-PEG-PLLA/PPZn composite films (**a**) without PPZn and with PPZn contents of (**b**) 0.5 %wt, (**c**) 1 %wt, (**d**) 2 %wt, and (**e**) 4 %wt after testing dimensional stability to heat at 80 °C under a 200 g load for 30 s.

**Figure 8 polymers-16-00975-f008:**
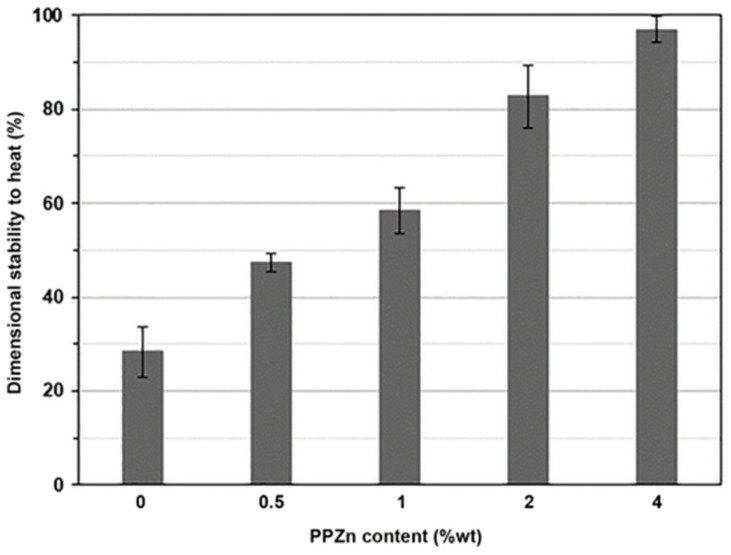
Dimensional stability to heat of PLLA-PEG-PLLA/PPZn composite films with various PPZn contents.

**Figure 9 polymers-16-00975-f009:**
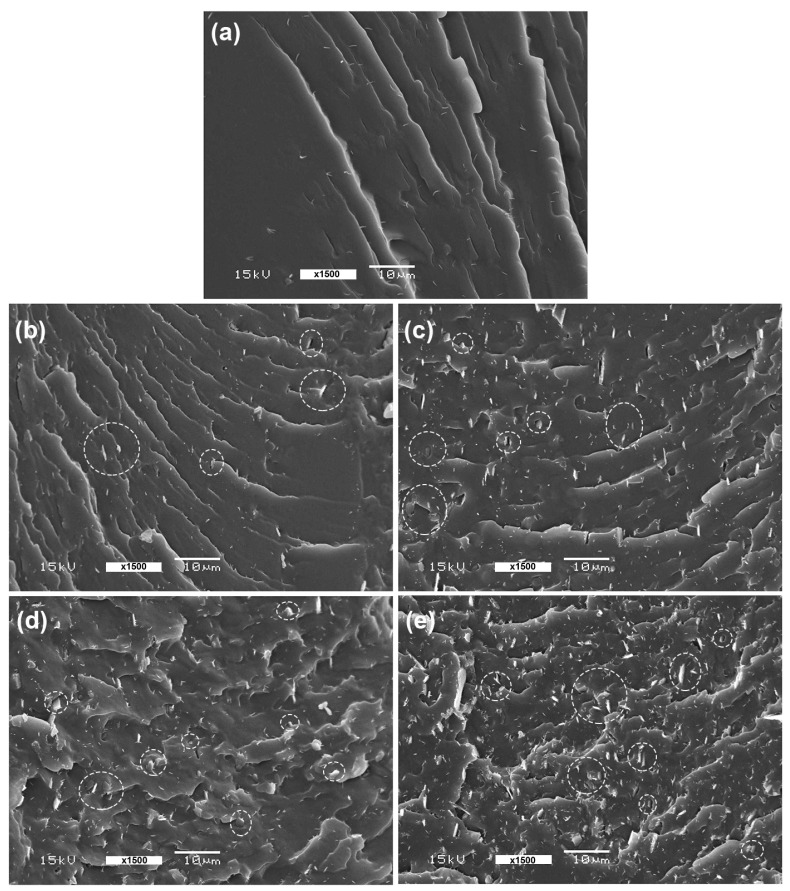
SEM images of cryo-fractured surfaces of PLLA-PEG-PLLA/PPZn composite films (**a**) without PPZn and with PPZn contents of (**b**) 0.5 %wt, (**c**) 1 %wt, (**d**) 2 %wt, and (**e**) 4 %wt (some PPZn particles were labeled by white circles; all bar scales = 10 µm).

**Figure 10 polymers-16-00975-f010:**
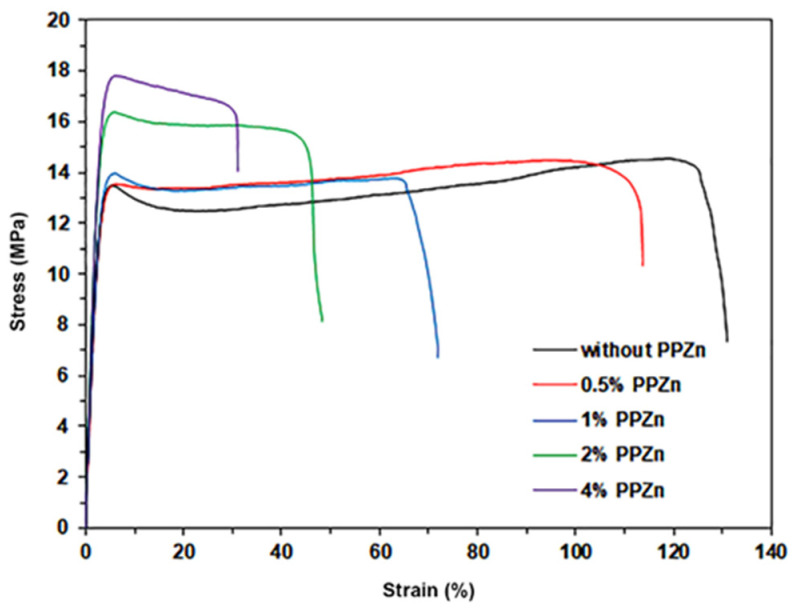
Tensile curves of PLLA-PEG-PLLA/PPZn composite films with various PPZn contents.

**Figure 11 polymers-16-00975-f011:**
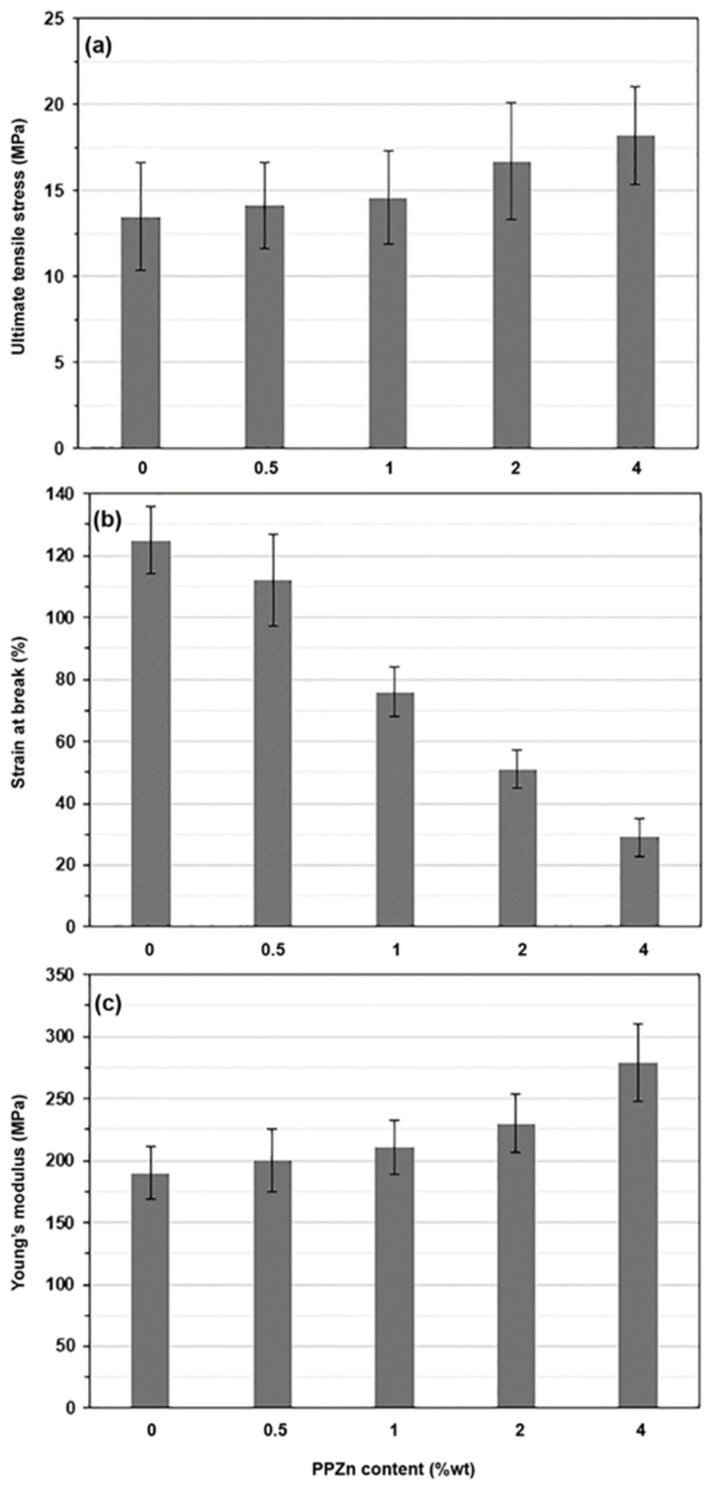
(**a**) Ultimate tensile stress, (**b**) strain at break, and (**c**) Young’s modulus of PLLA-PEG-PLLA/PPZn composite films with various PPZn contents from Table 4.

**Figure 12 polymers-16-00975-f012:**
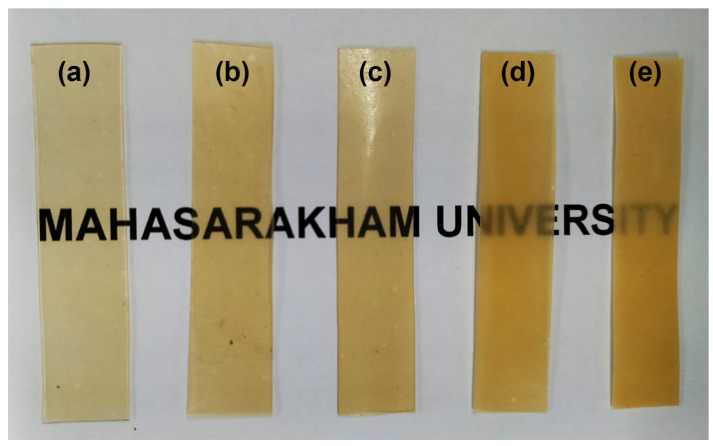
Photographs of PLLA-PEG-PLLA/PPZn composite films (**a**) without PPZn and with PPZn contents of (**b**) 0.5 %wt, (**c**) 1 %wt, (**d**) 2 %wt, and (**e**) 4 %wt.

**Table 1 polymers-16-00975-t001:** DSC results obtained from heating and cooling thermograms of PLLA-PEG-PLLA/PPZn composites.

PPZn Content (%wt)	*W_PLLA_* ^1^	*T_g_* ^2^ (°C)	*T_cc_* ^2^ (°C)	Δ*H_cc_* ^2^ (J/g)	*T_m_* ^2^ (°C)	Δ*H_m_* ^2^ (J/g)	*DSC-X_c_* ^2^ (%)	*T_c_* ^3^ (°C)
-	0.830	31	81	16.5	152	26.1	12.4	107
0.5	0.826	-	-	-	153	32.6	42.2	118
1	0.822	-	-	-	153	33.4	43.4	119
2	0.813	-	-	-	153	34.7	45.6	120
4	0.797	-	-	-	153	37.0	49.6	120

^1^ Weight fraction of PLLA (*W_PLLA_* of PLLA for PLLA-PEG-PLLA is 0.830 [24,36]). ^2^ Obtained from DSC heating thermograms. ^3^ Obtained from DSC cooling thermograms.

**Table 2 polymers-16-00975-t002:** Crystallization half-time (*t*_1/2_) and Avrami parameters (*n* and *k*) of PLLA-PEG-PLLA/PPZn composites obtained from isothermal curves at 120 °C.

PPZn Content (%wt)	*t*_1/2_ (min)	*n*	*k* (min^−k^)	*R* ^2^
-	2.18	3.8569	0.0208	0.9986
0.5	0.75	3.2228	2.2596	0.9968
1	0.69	3.2092	4.2982	0.9983
2	0.58	3.2012	4.6353	0.9955
4	0.47	2.5142	5.9607	0.9970

**Table 3 polymers-16-00975-t003:** Thermal decomposition of PLLA-PEG-PLLA/PPZn composites.

PPZn Content (%wt)	Residue Weight at 800 °C ^1^ (%)	*PLLA-T_d,max_*^2^ (°C)	*PEG-T_d,max_*^2^ (°C)
-	0.42	323	416
0.5	0.97	328	417
1	1.22	328	418
2	2.15	328	416
4	3.66	329	415

^1^ Obtained from TG thermograms. ^2^ Obtained from DTG thermograms.

**Table 4 polymers-16-00975-t004:** Tensile properties and opacity of PLLA-PEG-PLLA/PPZn composite films.

PPZn Content (%wt)	Ultimate Tensile Stress (MPa)	Strain at Break (%)	Young’s Modulus (Mpa)	Opacity (mm^−1^)
-	13.5 ± 3.1	125 ± 11	190 ± 21	0.402 ± 0.088
0.5	14.1 ± 2.5	112 ± 15	200 ± 25	0.754 ± 0.045
1	14.6 ± 2.7	76 ± 8	211 ± 22	0.936 ± 0.027
2	16.7 ± 3.4	51 ± 6	230 ± 24	1.960 ± 0.067
4	18.2 ± 2.8	29 ± 6	279 ± 31	3.829 ± 0.074

## Data Availability

Data are contained within the article.

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
