# Peer review of "Improvement in Crystallization, Thermal, and Mechanical Properties of Flexible Poly(L-lactide)-*b*-poly(ethylene glycol)-*b*-poly(L-lactide) Bioplastic with Zinc Phenylphosphate"

_polymers, 2024, doi:10.3390/polym16070975_

Round 1

Reviewer 1 Report

Comments and Suggestions for Authors

The authors presented a research article on the morphological, mechanical, thermal properties of Poly(L-lactide)-b-poly(ethylene glycol)-b-poly(L-lactide) (PLLA-PEG-PLLA) doped with zinc phenylphosphate.

PLLA is a brittle biopolymer with low strength, which prevents its widespread use. To improve its characteristics, the authors use mixing PLLA with polyethylene glycol. PEG acts as a plasticizer, but the PLLA/PEG mixture is not stable and phase separation is possible. Therefore, the introduction of PPZn seems justified in many aspects. Overall, the article looks like a complete study, but the clarity and detail could be improved. I would recommend publishing the article after revision and some additions, taking into account the following comments:

1.      The introduction could be improved by describing in more detail the need for plasticization of PLA and the use of polyethylene glycol. (line 45)

2.      In the Materials section:

The given SEM micrograph of PPZn should be supplemented by a graph of particle size distribution. The XRD plot of these particles should be presented here - not in supplementary materials, since the particles must be fully characterized. The fact that the particles have a size less than 5 microns is not indicative.

3.      DSC:

In Figures 2 and 3, we observe the 2nd heating and do not see the glass transition of the polymer chain and its change depending on the amount of added PPZn. There is no understanding of the mobility of the polymer chain because the material has been preheated. There is a characteristic shoulder in Tm, which indicates the melting of additional crystals. This is not described anywhere. Table 1 does not include ∆Hm, ∆Hcc, etc. Table 1 does not include ∆Hm, ∆Hcc, etc. This data must be added.

4.      XRD:

It is necessary for better visualization to space the curves in the Figure and add PPZn XRD. The intensity must be indicated in the a.u.

5.      Also, please clarify the huge difference between the crystallinity data from DSC and XRD. The data should be close in value. According to your data: 28.1% degree of crystallinity for the addition of 4% PPZn for 28.1% XRD and 49.6% for DSC. And for other samples too. Explain.

6.      DMA: Since in this study the samples were preheated, it is difficult to judge the reliability of the data. When comparing the data from DMA and Emod mechanical tests, they should coincide or have very close values. Apparently, the samples were poorly secured and crawled out of the fastening (sagged), and therefore the Emod at DMA was lower. Tan a values are not available, although this is necessary data for assessing the effect of crystallinity on the material.

7.      Mechanical tests: There is no data on the elongation of the obtained materials. In the Dimentional stability section, significant changes are visible and a reduction in size of more than 2 times is visible as a result of increasing the amount of additive. I consider it necessary to indicate the exact dimensions of the samples for mechanical tests and provide data on elongation (without heating).

8.      TGA: In Figure 4, show an enlarged area from 2000С to 6700С and separately from 2000С to 4500С. Also, superimpose the DTG data on each other to assess the changes occurring in the samples and correlate them with the data in Table 3.

9.      Please add references to the literature in the discussion of XRD and DMA, TGA and DTG effects.

10.   Have FT-IR spectroscopy studies been carried out?

11.   Expand the list of references

Comments on the Quality of English Language

The article is written quite well. Moderate English editing required

Author Response

Reviewer # 1_Round 1

Manuscript ID: polymers-2910230

Title: Improvement in crystallization, thermal, and mechanical properties of flexible poly(L-lactide)-b-poly(ethylene glycol)-b-poly(L-lactide) bioplastic with zinc phenylphosphate

Authors: Kansiri Pakkethati, Prasong Srihanam, Apirada Manphae, Wuttipong Rungseesantivanon, Natcha Prakymoramas, Pham Ngoc Lan, and Yodthong Baimark

Reviewer # 1

PLLA is a brittle biopolymer with low strength, which prevents its widespread use. To improve its characteristics, the authors use mixing PLLA with polyethylene glycol. PEG acts as a plasticizer, but the PLLA/PEG mixture is not stable and phase separation is possible. Therefore, the introduction of PPZn seems justified in many aspects. Overall, the article looks like a complete study, but the clarity and detail could be improved. I would recommend publishing the article after revision and some additions, taking into account the following comments:

Authors: The authors would like to sincerely thank the reviewer for the time that the reviewer spent reading the paper and their perceptive comments. All the comments have been used to improve the paper. A detailed point-by-point set of responses to the reviewer inputs is provided. All corrections are highlighted in red.

  1. The introduction could be improved by describing in more detail the need for plasticization of PLA and the use of polyethylene glycol. (line 45).

Authors: More detail about plasticization of PLLA with poly(ethylene glycol) (PEG) has been provided on P. 1-2, lines 42-50 of the revised manuscript as follows.

In order to increase PLLA’s flexibility by increasing its chain mobility, a wide range of plasticizers have been extensively blended with PLLA [14-16]. The selection of a good plasticizer for PLLA has considered a number of factors, including plasticizer efficiency, non-toxicity, good miscibility, and durability. Among all the plasticizers, poly(ethylene glycol) (PEG) was the most effective for PLLA due to it being non-toxic, good miscibility, and ability to accelerate PLLA’s hydrolytic degradation [16]. Nevertheless, the primary drawback has been identified as phase separation and PEG migration from PLLA matrices [17-19], which has a direct impact on the stability and durability of the PLLA/PEG blends throughout storage and use [20,21].

  1. In the Materials section: The given SEM micrograph of PPZn should be supplemented by a graph of particle size distribution. The XRD plot of these particles should be presented here - not in supplementary materials, since the particles must be fully characterized. The fact that the particles have a size less than 5 microns is not indicative.

Authors: A graph of particle size distribution of PPZn powder was presented in Figure 1(b) of the revised manuscript. The SEM image, TG thermogram, and XRD pattern of PPZn are presented in Figures 1(a), 1(c), and 1(d), respectively of the revised manuscript. Characteristics of PPZn are now described on P. 3, lines 97-104 of the revised manuscript as follows.

Figure 1(a) shows a SEM image of the obtained PPZn powder. Average particle size of 100 PPZn particles determined from SEM image using an ImageJ program was 4.12 ± 1.60 µm. The particle size distribution of PPZn powder is presented in Figure 1(b). Thermal decomposition of PPZn powder determined from thermogravimetric (TG) thermogram was in range 550 °C - 700 °C as shown in Figure 1(c). The residue weight at 800 °C of PPZn powder was about 73%. The XRD pattern of PPZn powder is illustrated in Figure 1(d) with XRD peaks at 6.5°, 12.6°, and 18.5°.

  1. DSC: In Figures 2 and 3, we observe the 2nd heating and do not see the glass transition of the polymer chain and its change depending on the amount of added PPZn. There is no understanding of the mobility of the polymer chain because the material has been preheated. There is a characteristic shoulder in Tm, which indicates the melting of additional crystals. This is not described anywhere. Table 1 does not include ∆Hm, ∆Hcc, etc. Table 1 does not include ∆Hm, ∆Hcc, etc. This data must be added.

Authors: PLLA-PEG-PLLA/PPZn composites had higher crystallinity than pure PLLA-PEG-PLLA after quenching from 200 °C to 0 °C. More detail on restriction of chain mobility in amorphous phases of PLLA-PEG-PLLA composites from crystalline phases is described on P. 5, lines 196-200 of the revised manuscript as follows.

This may be due to the glassy-to-rubbery transition in amorphous regions of PLLA end-blocks after quenching from 200 °C to 0 °C being difficult to detect by DSC for PLLA-PEG-PLLA composites because the free volume of the polymer chains was reduced for high-crystallinity polymers, which leads to restricted the motion of polymer chains in amorphous regions [42].

          Characteristic of shoulder Tm peak, which indicates the melting of additional crystals has been described on P. 5, lines 202-205 of the revised manuscript as follows.

It should be noted that lower-temperature shoulder Tm peaks were also detected when the PPZn was incorporated, which suggested that imperfect crystals of PLLA end-blocks were formed [29].

          The ∆Hm and ∆Hcc values have been added in Table 1 on P. 6 of the revised manuscript as follows.

Table 1. DSC results obtained from heating and cooling thermograms of PLLA-PEG-PLLA/PPZn composites.

PPZn content

(%wt)

Tg 1

(°C)

Tcc 1

(°C)

DHcc 1

(J/g)

Tm 1

(°C)

DHm 1

(J/g)

DSC-Xc 1

(%)

Tc 2

(°C)

-

0.5

1

2

4

31

-

-

-

-

81

-

-

-

-

16.5

-

-

-

-

152

153

153

153

153

26.1

32.6

33.4

34.7

37.0

12.4

42.2

43.4

45.6

49.6

107

118

119

120

120

1 Obtained from DSC heating thermograms.

2 Obtained from DSC cooling thermograms.

  1. XRD: It is necessary for better visualization to space the curves in the Figure and add PPZn XRD. The intensity must be indicated in the a.u.

Authors: The space of the XRD curves has been increased as shown in Figure 5 on P. 9-10 of the revised manuscript as follows.

Figure 5. XRD patterns of PLLA-PEG-PLLA/PPZn composites (a) without PPZn and with PPZn contents of (b) 0.5 %wt, (c) 1 %wt, (d) 2 %wt, and (e) 4 %wt.

     The XRD pattern of PPZn powder has been added as Figure 1(d) on P. 3 of the revised manuscript as follows.

Figure 1. (a) SEM image, (b) particle size distribution, (c) TG thermogram, and (d) XRD pattern of PPZn powder.

  1. Also, please clarify the huge difference between the crystallinity data from DSC and XRD. The data should be close in value. According to your data: 28.1% degree of crystallinity for the addition of 4% PPZn for 28.1% XRD and 49.6% for DSC. And for other samples too. Explain.

Authors: Explanation of the difference between the DSC-Xc and XRD-Xc values has described on P. 9, lines 317-320 of the revised manuscript as follows.

The difference between the DSC-Xc and XRD-Xc values could be due to the possible crystallization of the samples during DSC heating scans [27,53]. In addition, the film samples used in the XRD test were compressed and cooled to limit the mobility of the polymer chains for crystallization.

  1. DMA: Since in this study the samples were preheated, it is difficult to judge the reliability of the data.

When comparing the data from DMA and Emod mechanical tests, they should coincide or have very close values. Apparently, the samples were poorly secured and crawled out of the fastening (sagged), and therefore the Emod at DMA was lower. Tan a values are not available, although this is necessary data for assessing the effect of crystallinity on the material.

Authors: In this work, the storage modulus in Figure 6 (P. 10) from DMA analysis was used to consider the stiffness of film samples during DMA heating scan with tension mode. The storage modulus and stiffness of film sample was directly related to its heat resistant properties as previously reported in the literature [ref. nos. 27,40,43,45,54-59].

  1. Mechanical tests: There is no data on the elongation of the obtained materials. In the Dimentional stability section, significant changes are visible and a reduction in size of more than 2 times is visible as a result of increasing the amount of additive. I consider it necessary to indicate the exact dimensions of the samples for mechanical tests and provide data on elongation (without heating).

Authors: Tensile properties including ultimate tensile stress, strain at break, and Young’s modulus were used to determine the mechanical properties of the film samples that are reported in section 3.7 on P. 12-14 of the revised manuscript.

  1. TGA: In Figure 4, show an enlarged area from 200°С to 670°С and separately from 200°С to 450°С. Also, superimpose the DTG data on each other to assess the changes occurring in the samples and correlate them with the data in Table 3.

Authors: TG and DTG thermograms of the samples were enlarged as shown in Figure 4 on P. 8 of the revised manuscript as follows.

Figure 4. (a) TG and (b) DTG thermograms of PLLA-PEG-PLLA/PPZn composites with various PPZn contents as well as (c) expanded TG and (d) expanded DTG thermograms in decomposition region of PLLA end-blocks.

  1. Please add references to the literature in the discussion of XRD and DMA, TGA and DTG effects.

Authors: More references have been added for discussion on XRD (ref. nos. 27 and 53) on P. 9, lines 317-320 as follows.

  The difference between the DSC-Xc and XRD-Xc values could be due to the possible crystallization of the samples during DSC heating scans [27,53]. In addition, the film samples used in the XRD test were compressed and cooled to limit the mobility of the polymer chains for crystallization.

More references have added to support the discussion on DMA (ref. nos. 19, 27,40,43,45,54-59) on P. 10, lines 326-350.

More references have added to support the discussion on TGA (ref. nos. 51 and 52) on P. 8, lines 286-289 and on P. 9, lines 296-297 of the revised manuscript as follows.

This may be due to a good heat transfer from the PLLA-PEG-PLLA matrices to the high heat-stability-PPZn that improved the thermal stability of the PLLA-PEG-PLLA attributed to good phase adhesion between matrix and filler [51].

In addition, Yang et al. have reported that the thermal stability of PLLA was improved with increasing crystallinity of PLLA [52].

  1. Have FT-IR spectroscopy studies been carried out?

Authors: We apologize for the lack of FTIR analysis of the film samples. Due to the unavailability of our instruments at this time.

  1. Expand the list of references.

Authors: List of references has been expanded from 48 to 61 as listed on P. 16-19 of the revised manuscript.

Concluding Remarks

We hope that our responses answer the reviewers’ comments to their satisfaction and that the revisions that have been made to the paper enhance its clarity for the benefit of the reader.

Yours Faithfully,

The Authors

Reviewer 2 Report

Comments and Suggestions for Authors

The manuscript entitled "Improvement in Crystallization, Thermal, and Mechanical Properties of Flexible Poly(L-lactide)-b-poly(ethylene glycol)-b-poly(L-lactide) Bioplastic with Zinc phenylphosphate" focused on the PLA-based film properties evaluation. The zinc-based nucleating agents we already used for the modification of poly(lactic acid), which means that the topic of the presented study is not novel. However, considering that the results of the work may constitute an interesting reference point for future research, I believe that the material has publication potential. Before publication, some major corrections should be made:

1. The article covers the topic of foil processing; however, the authors do not refer in their analyses to the assessment of the impact of the extrusion process conditions, which in the case of PLA products has a very large impact on the properties. I suggest supplementing the Introduction section with information on this topic.

2. The tests should present the appearance of the foil so that a qualitative assessment of the prepared materials is possible.

3. Another aspect where the reference is missing in the work is the transparency of the sample, I suggest performing Haze measurements

4. The increase in crystallinity has a very significant impact on the barrier properties of the foil, therefore the work should include this issue

Author Response

Reviewer # 2_Round 1

Manuscript ID: polymers-2910230

Title: Improvement in crystallization, thermal, and mechanical properties of flexible poly(L-lactide)-b-poly(ethylene glycol)-b-poly(L-lactide) bioplastic with zinc phenylphosphate

Authors: Kansiri Pakkethati, Prasong Srihanam, Apirada Manphae, Wuttipong Rungseesantivanon, Natcha Prakymoramas, Pham Ngoc Lan, and Yodthong Baimark

Reviewer # 2

The manuscript entitled "Improvement in Crystallization, Thermal, and Mechanical Properties of Flexible Poly(L-lactide)-b-poly(ethylene glycol)-b-poly(L-lactide) Bioplastic with Zinc phenylphosphate" focused on the PLA-based film properties evaluation. The zinc-based nucleating agents we already used for the modification of poly(lactic acid), which means that the topic of the presented study is not novel. However, considering that the results of the work may constitute an interesting reference point for future research, I believe that the material has publication potential. Before publication, some major corrections should be made:

Authors: The authors would like to sincerely thank the reviewer for the time that the reviewer spent reading the paper and for their perceptive comments. All the comments have been used to improve the paper. A detailed point-by-point set of responses to the reviewer inputs is provided. All corrections are highlighted in red.

  1. The article covers the topic of foil processing; however, the authors do not refer in their analyses to the assessment of the impact of the extrusion process conditions, which in the case of PLA products has a very large impact on the properties. I suggest supplementing the Introduction section with information on this topic.

Authors: The impact of the PLLA processing on its crystallization has been added in the Introduction on P. 2, lines 65-70 of the revised manuscript as follows.

Depending on thermal history of PLLA, it can be either amorphous or semicrystalline. PLLA will become highly amorphous upon quenching it from the melt phase (for example, during the extrusion and injection processes) [26]. Addition of a nucleating agent to the PLLA is an effective method for improving the crystallization of PLLA during its processing. Crystallization at high temperature can occur because the surface free energy barrier for nucleation is lowered upon cooling.

  1. The tests should present the appearance of the foil so that a qualitative assessment of the prepared materials is possible.

Authors: The appearance of the films is illustrated in Figure 12 on P. 15 of the revised manuscript as follows.

Figure 12. Photographs of PLLA-PEG-PLLA/PPZn composite films (a) without PPZn and with PPZn contents of (b) 0.5 %wt, (c) 1 %wt, (d) 2 %wt, and (e) 4 %wt.

  1. Another aspect where the reference is missing in the work is the transparency of the sample, I suggest performing Haze measurements.

Authors: Opacity of film samples has been analyzed and reported in the revised manuscript. The method for measurement of film’s opacity is now described on P. 4, lines 178-183 of the revised manuscript as follows.

The film’s opacity was determined using a Thermo Scientific Genesys 20 visible spectrophotometer (Loughborough, UK) and was calculated with the following equation [41].

                            Opacity (mm-1) = A600/X                              (3)

                             where A600 is the absorbance of film at 600 nm and X is the thickness of film sample (mm).

The results of film’s opacity are reported in Table 4 on P. 13 of the revised manuscript as follows.

Table 4. Tensile properties and opacity of PLLA-PEG-PLLA/PPZn composite films.

PPZn content

(%wt)

Ultimate tensile stress (MPa)

Strain at break

(%)

Young’s modulus

(MPa)

Opacity

(mm-1)

-

0.5

1

2

4

13.5 ± 3.1

14.1 ± 2.5

14.6 ± 2.7

16.7 ± 3.4

18.2 ± 2.8

125 ± 11

112 ± 15

76 ± 8

51 ± 6

29 ± 6

190 ± 21

200 ± 25

211 ± 22

230 ± 24

279 ± 31

0.402 ± 0.088

0.754 ± 0.045

0.936 ± 0.027

1.960 ± 0.067

3.829 ± 0.074

The results of film’s opacity are described on P. 14, lines 438-444 of the revised manuscript as follows.

3.8. Film’s opacity

The opacity of pure PLLA-PEG-PLLA film calculated from equation (3) was 0.402 mm-1 as also reported in Table 4. It was found that the opacity of film samples increased when the added PPZn content was increased. The composite films had higher opacity compared with the pure PLLA-PEG-PLLA film as shown in Figure 12. However, the words covered by the composite films were still clearly visible, and were legible. It has been reviewed that the polymer film’s opacity increased as the crystallinity of polymers increased [61].

  1. The increase in crystallinity has a very significant impact on the barrier properties of the foil, therefore the work should include this issue.

Authors: Yes, barrier properties of PLLA are dependent on its crystallinity. We apologize for the analysis of the barrier properties of the film samples. Due to the unavailability of our instruments at this time.

Concluding Remarks

We hope that our responses answer the reviewers’ comments to their satisfaction and that the revisions that have been made to the paper enhance its clarity for the benefit of the reader.

Yours Faithfully,

The Authors

Round 2

Reviewer 1 Report

Comments and Suggestions for Authors

The author substantially improved the manuscript in accordance with the reviewer’s suggestions, however, some issues are still there in data presentation and discussion of obtained data. After addressing these points, the paper could be accepted for publication. Particular comments to the revised manuscript are given below:

1. In DCS experiments, what was the cooling rate during quenching to 0 oC after holding at 200 oC for 3 minutes? And how it is related to the cooling rate during PLLA film preparation using compression molding? This data should be added to the Materials and Methods section. Why it is necessary to preheat the samples and hold them at 200 oC before DSC heating scan? After quenching, the samples used in DSC heating scan could be not exactly the same as after compression molding.

It was shown by the authors in isothermal crystallization experiments that PLLA crystallization rate is highly increased with addition of PPZn, as compared to pure PLLA sample. If these two aforementioned cooling rates (during molding and in DSC experiment) were substantially different, the crystallization conditions of PPLA during film preparation and during DSC experiment could be also different. This could lead to the discrepancy in the degree of crystallinity values obtained from DSC and XRD. Therefore, it is suggested to provide DSC heating scans for intact prepared PLLA films (without preheating to 200 oC), since exactly the same samples were used for other characterization methods (XRD, DMA, mechanical tests).

Also, the isothermal crystallization curves for the PLLA samples with 1% and 2% PPZn content almost coincide, but the resulting Avrami parameters (n and k values) obtained after fitting the experimental data with the Avrami equation are significantly different. The error values for the n and k values obtained as the result of fitting should be provided in Table 2.

In addition, the exact weight fraction of the PLLA should be indicated for all studied samples in Table 1.

2. XRD presentation is still confusing. It seems from the XRD pattern in Figure 5 that pure PLLA-PEG-PLLA film is almost amorphous. It raises some doubts for the estimated XRD-Xc value of pure PLLA-PEG-PLLA film. Please give the XRD pattern for this film in separate graph or in the inset.

Also, the explanation for the observed discrepancy in the Xc values obtained from DSC and XRD is somewhat controversial. The authors wrote:

The difference between the DSC-Xc and XRD-Xc values could be due to the possible crystallization of the samples during DSC heating scans [27,53].

But this is exactly the process of cold crystallization, which was not observed in DSC scans for the PLLA-PEG-PLLA/PPZn composites according to the presented results. Or this process occurs in much broader temperature range as compared to the pure PLLA-PEG-PLLA film and should be taken into account in the analysis of DSC data.

The discussion of the obtained experimental Xc values from XRD data should be improved and brought into line with the discussion of DSC data.

Comments on the Quality of English Language

English could be improved. There are inconsistent proposals in some places.

Author Response

Reviewer # 1_Round 2

Manuscript ID: polymers-2910230

Title: Improvement in crystallization, thermal, and mechanical properties of flexible poly(L-lactide)-b-poly(ethylene glycol)-b-poly(L-lactide) bioplastic with zinc phenylphosphate

Authors: Kansiri Pakkethati, Prasong Srihanam, Apirada Manphae, Wuttipong Rungseesantivanon, Natcha Prakymoramas, Pham Ngoc Lan, and Yodthong Baimark

Reviewer 1:

The author substantially improved the manuscript in accordance with the reviewer’s suggestions, however, some issues are still there in data presentation and discussion of obtained data. After addressing these points, the paper could be accepted for publication. Particular comments to the revised manuscript are given below:

Authors: The authors would like to sincerely thank the reviewer for the time that the reviewer spent reading the paper and for their perceptive comments. All the comments have been used to improve the paper. A detailed point-by-point set of responses to the reviewer inputs is provided. All corrections are highlighted in red.

  1. In DCS experiments, what was the cooling rate during quenching to 0 oC after holding at 200 oC for 3 minutes? This data should be added to the Materials and Methods section. Why it is necessary to preheat the samples and hold them at 200 oC before DSC heating scan?

Authors: For DSC test, the cooling rate during quenching to 0 oC after holding at 200 oC for 3 minutes was 100 °C/min. The preheat the samples and hold them at 200 oC before DSC heating scan used to remove their thermal history. This has described on P. 3, lines 124-126 of the revised manuscript as follows.

For non-isothermal analysis, the samples were maintained at 200 °C for 3 min to erase the previous thermal history before quickly quenching to 0 °C with cooling rate of 100 °C/min.

  1. And how it is related to the cooling rate during PLLA film preparation using compression molding? After quenching, the samples used in DSC heating scan could be not exactly the same as after compression molding.

Authors: Yes, cooling conditions between DSC quenching and compression cooling are different according to reviewer’s comment. This was used to discuss the difference between the DSC-Xc and XRD-Xc values on P. 9, lines 318-320 of the revised manuscript as follows.

The difference between the DSC-Xc and XRD-Xc values could be due to the differences in crystallization conditions during the cooling process of the samples [27,53].

  1. It was shown by the authors in isothermal crystallization experiments that PLLA crystallization rate is highly increased with addition of PPZn, as compared to pure PLLA sample. If these two aforementioned cooling rates (during molding and in DSC experiment) were substantially different, the crystallization conditions of PLLA during film preparation and during DSC experiment could be also different. This could lead to the discrepancy in the degree of crystallinity values obtained from DSC and XRD. Therefore, it is suggested to provide DSC heating scans for intact prepared PLLA films (without preheating to 200 oC), since exactly the same samples were used for other characterization methods (XRD, DMA, mechanical tests).

Authors: Yes, difference in cooling conditions during DSC tests and compression molding induced on the difference between the DSC-Xc and XRD-Xc values. This is discussed on P. 9, lines 318-320 of the revised manuscript as follows.

The difference between the DSC-Xc and XRD-Xc values could be due to the differences in crystallization conditions during the cooling process of the samples [27,53].

The non-isothermal and isothermal DSC scans was used to determine the crystallization properties of the pure PLLA-PEG-PLLA compared with the composites after thermal history removing. Therefore, crystallinity from DSC can not compared with crystallinity from XRD because of difference in cooling condition according to reviewer’s comment. The objective of this work is study on effect of PPZn on crystallization properties of PLLA-PEG-PLLA. It was found that the trend of the crystallinity of samples from DSC and XRD tends to increase as the PPZn content increases. The results from DSC and XRD support each other that PPZn is a crystallizing agent for PLLA-PEG-PLLA as described on P. 9, lines 316-318 of the revised manuscript as follows.

The XRD results supported the conclusion that the PPZn acted as a nucleating agent according to the above DSC results.

  1. Also, the isothermal crystallization curves for the PLLA samples with 1% and 2% PPZn content almost coincide, but the resulting Avrami parameters (n and k values) obtained after fitting the experimental data with the Avrami equation are significantly different. The error values for the n and k values obtained as the result of fitting should be provided in Table 2.

In addition, the exact weight fraction of the PLLA should be indicated for all studied samples in Table 1.

Authors: Yes, the isothermal crystallization curves for the PLLA samples with 1% and 2% PPZn content almost coincide. Then n and k values of 1% and 2% PPZn content are nearly values as reported in Table 2. The n value tends to decrease and k value tends to increase as the amount of PPZn increases. For isothermal DSC test, the sample was scanned for one scan.

The exact values of weight fraction of the PLLA (WPLLA) were added in Table 1 on P. 6 of the revised manuscript as follows.

Table 1. DSC results obtained from heating and cooling thermograms of PLLA-PEG-PLLA/PPZn composites.

PPZn content

(%wt)

WPLLA1

Tg 2

(°C)

Tcc 2

(°C)

DHcc 2

(J/g)

Tm 2

(°C)

DHm 2

(J/g)

DSC-Xc 2

(%)

Tc 3

(°C)

-

0.5

1

2

4

0.830

0.826

0.822

0.813

0.797

31

-

-

-

-

81

-

-

-

-

16.5

-

-

-

-

152

153

153

153

153

26.1

32.6

33.4

34.7

37.0

12.4

42.2

43.4

45.6

49.6

107

118

119

120

120

1 Weight fraction of PLLA (WPLLA of PLLA for PLLA-PEG-PLLA is 0.830 [24,36]).

2 Obtained from DSC heating thermograms.

                                                                                               3 Obtained from DSC cooling thermograms.

  1. XRD presentation is still confusing. It seems from the XRD pattern in Figure 5 that pure PLLA-PEG-PLLA film is almost amorphous. It raises some doubts for the estimated XRD-Xc value of pure PLLA-PEG-PLLA film. Please give the XRD pattern for this film in separate graph or in the inset.

Authors: The XRD pattern of pure PLLA-PEG-PLLA film was enlarged as shown in Figure 5 on P. 9 of the revised manuscript as follows.

Figure 5. XRD patterns of PLLA-PEG-PLLA/PPZn composites (a) without PPZn and with PPZn contents of (b) 0.5 %wt, (c) 1 %wt, (d) 2 %wt, and (e) 4 %wt.

  1. Also, the explanation for the observed discrepancy in the Xc values obtained from DSC and XRD is somewhat controversial. The authors wrote:

The difference between the DSC-Xc and XRD-Xc values could be due to the possible crystallization of the samples during DSC heating scans [27,53].

But this is exactly the process of cold crystallization, which was not observed in DSC scans for the PLLA-PEG-PLLA/PPZn composites according to the presented results. Or this process occurs in much broader temperature range as compared to the pure PLLA-PEG-PLLA film and should be taken into account in the analysis of DSC data.

The discussion of the obtained experimental Xc values from XRD data should be improved and brought into line with the discussion of DSC data.

Authors: This sentence was re-written on P. 9, lines 318-320 of the revised manuscript as follows.

The difference between the DSC-Xc and XRD-Xc values could be due to the differences in crystallization conditions during the cooling process of the samples [27,53].

Concluding Remarks

We hope that our responses answer the reviewers’ comments to their satisfaction and that the revisions that have been made to the paper enhance its clarity for the benefit of the reader.

Yours Faithfully,

The Authors

Reviewer 2 Report

Comments and Suggestions for Authors

The newest version of the article was strongly improved, and the author has answered most of the comments. In my opinion, the presented version of the manuscript can be processed for publication. 

Author Response

Reviewer # 2_Round 2

Manuscript ID: polymers-2910230

Title: Improvement in crystallization, thermal, and mechanical properties of flexible poly(L-lactide)-b-poly(ethylene glycol)-b-poly(L-lactide) bioplastic with zinc phenylphosphate

Authors: Kansiri Pakkethati, Prasong Srihanam, Apirada Manphae, Wuttipong Rungseesantivanon, Natcha Prakymoramas, Pham Ngoc Lan, and Yodthong Baimark

Reviewer # 2

The newest version of the article was strongly improved, and the author has answered most of the comments. In my opinion, the presented version of the manuscript can be processed for publication. 

Authors: The authors would like to sincerely thank the reviewer for the time that the reviewer spent reading the paper.

Yours Faithfully,

The Authors
